# Genome Identification of the Tea Plant (*Camellia sinensis*) ASMT Gene Family and Its Expression Analysis under Abiotic Stress

**DOI:** 10.3390/genes14020409

**Published:** 2023-02-04

**Authors:** Fangfang Xu, Wenxiang Liu, Hui Wang, Pravej Alam, Wei Zheng, Mohammad Faizan

**Affiliations:** 1College of Forestry, Xinyang Agriculture and Forestry University, Xinyang 464000, China; 2Department of Biology, College of Science and Humanities, Prince Sattam Bin Abdulaziz University, Alkharj 11942, Saudi Arabia; 3Botany Section, School of Sciences, Maulana Azad National Urdu University, Hyderabad 500032, India

**Keywords:** abiotic stress, gene expression, *Camellia sinensis*, ASMT gene, melatonin

## Abstract

The tea plant (*Camellia sinensis* (L.) O. Ktze) is an important cash crop grown worldwide. It is often subjected to environmental stresses that influence the quality and yield of its leaves. Acetylserotonin-O-methyltransferase (ASMT) is a key enzyme in melatonin biosynthesis, and it plays a critical role in plant stress responses. In this paper, a total of 20 *ASMT* genes were identified in tea plants and classified into three subfamilies based on a phylogenetic clustering analysis. The genes were unevenly distributed on seven chromosomes; two pairs of genes showed fragment duplication. A gene sequence analysis showed that the structures of the *ASMT* genes in the tea plants were highly conserved and that the gene structures and motif distributions slightly differed among the different subfamily members. A transcriptome analysis showed that most *CsASMT* genes did not respond to drought and cold stresses, and a qRT-PCR analysis showed that *CsASMT08*, *CsASMT09, CsASMT10,* and *CsASMT20* significantly responded to drought and low-temperature stresses; in particular, *CsASMT08* and *CsASMT10* were highly expressed under low-temperature stress and negatively regulated in response to drought stress. A combined analysis revealed that *CsASMT08* and *CsASMT10* were highly expressed and that their expressions differed before and after treatment, which indicates that they are potential regulators of abiotic stress resistance in the tea plant. Our results can facilitate further studies on the functional properties of *CsASMT* genes in melatonin synthesis and abiotic stress in the tea plant.

## 1. Introduction

Melatonin (N-acetyl-5-methoxytryptamine), a molecule with multiple functions in plant physiological responses, plays an important role in the regulation of stress tolerance in various plant species, including wheat and tobacco [1,2]. N-acetyl-5-hydroxytryptamine-methyltransferase (ASMT) is a key enzyme in melatonin biosynthesis, and it not only affects melatonin synthesis but also plays a crucial role in antioxidation [3,4,5,6] and abiotic stress responses [7,8,9,10]. For example, melatonin has been found to significantly enhance drought tolerance in plants by regulating the mitochondrial synthesis of melatonin [11,12]. Low doses of melatonin in rice have also been found to increase its antioxidant enzyme activity and non-enzymatic antioxidant levels, thus alleviating cold-stress-induced photosynthesis [13]. Melatonin is widely present in various plants, animals, and fungi; it is involved in various biological activities, and it is important for hormonal signaling and emergency responses [14,15].

Melatonin synthesis in plants involves four enzymes: tryptophan decarboxylase (TDC), 5-hydroxytryptamine-N-acetyltransferase (SNAT), tryptamine 5-hydroxylase (T5H), and acetylserotonin-O-methyltransferase (ASMT) [16,17]. Not only is ASMT a key enzyme in melatonin biosynthesis essential for regulating plant melatonin content, but it also plays a crucial role in the oxidative and abiotic stress response [3,4,5,6] and abiotic stress response [7,8,9,10], which is closely related to the regulation of melatonin levels by ASMT. Therefore, the systematic identification and analysis of the ASMT gene family members in tea plants are essential for understanding how the members of this family participate in abiotic stress regulation.

Plants live in complex and variable environments, and abiotic stresses, such as drought and cold stresses, cause damage to plant cells, destroy the cell structure, and reduce enzyme activation, as well as affecting their growth and development and influencing their spatial distribution and yield, thus fundamentally threatening production security [18]. In plants, melatonin acts as a protective agent and plays an important role in regulating plant tolerance to abiotic stresses. To date, the whole-genome identification of *ASMT* genes has been carried out in various plants, such as tomato [17] pepper [19], apple [20], walnut [21], and wild mulberry [22], and related genes have been shown to play important roles in abiotic stress responses, pathogen induction, and the melatonin synthesis pathway. In addition, previous studies have demonstrated that melatonin is involved in plant abiotic stress responses and that acetylserotonin-O-methyltransferase can methylate N-ethyl-5-hydroxytryptamine to form melatonin [23]. Many plants are deficient in ASMT homologs, which leads to impaired melatonin synthesis and reduced plant resilience to stress; thus, ASMT is essential for the regulation of stress tolerance. Few studies have been carried out on the functional validation of *ASMT* related genes in plants, and these have only investigated individual plants, such as rice and apple. The overexpression of *ASMT1*, *ASMT2*, and *ASMT3* in rice can enhance its enzymatic activity and drought tolerance [24], whereas the cloning and overexpression of the apple *MzASMT* gene in *Arabidopsis thaliana* lead to increases in melatonin levels and drought tolerance [25]. Thus, these studies suggest that *ASMT* genes are essential for the regulation of plant growth and stress responses.

Tea is a nutritious aromatic beverage and an economically valuable product. The quality and yield of tea plants decrease considerably under environmental stress. As *ASMT* genes play non-negligible roles in plant stress resistance, in the present study, we aimed to improve our understanding of the relevant functions of the *ASMT* genes in tea plants under abiotic stresses. We identified and classified the ASMT gene family members from the whole genome of tea plants based on bioinformatics approaches and chromosomal locations, gene duplication events, phylogenetic relationships, gene structures, conserved structural domains, and cis-elements. In addition, we investigated their expression profiles in different tissues under environmental stresses (cold and drought). Our findings provide a theoretical basis for further studies on the function of the ASMT gene family, as well as for understanding the molecular mechanism by which melatonin regulates stress tolerance in tea plants.

## 2. Materials and Methods

### 2.1. Material Treatment

The experimental materials originated from the teaching practice tea garden of Xinyang Agriculture and Forestry College, and two-year-old ‘Fuding Dabaicha’ tea plants from the same seedbed (free of pests and diseases and about 20 cm in height) were selected and transplanted into pots (Danish peat soil). The artificial climate chamber growth conditions were set as follows: a temperature of 22 ± 2 °C, humidity of 65%, untreated plants as control (CK), simulated drought (20% PEG 6000), and low-temperature (4 °C) treatment. The tea plants were watered with 20% PEG-6000 (500 mL), and one bud and three leaves were collected after drought treatment (0, 0.5, 1, 2, 4, and 8 d) and low-temperature treatment (0, 0.5, 1, 4, and 8 d). Three biological replicates were maintained for each treatment, and the samples were stored at −80 °C after quick freezing in liquid nitrogen.

### 2.2. Genome-Wide Identification of ASMT Proteins in C. sinensis

The ASMT structural protein (AK069308) model was downloaded from the Rice Protein Structure Database (http://structure.rice.dna.affrc.go.jp/ (accessed on 9 January 2022)) for a BLASTP search against the tea plant genome using an E-value < 1.0 × 10^−10^. Rice-related acetylserotonin-O-methyltransferase structural proteins, such as LOC_Os09g17560, LOC_Os10g02880, and LOC_Os10g02840 [26], were also used in the sequence query of CSS (cv. Shuchazao), with an E-value < 1.0 × 10^−10^. The ASMT Hidden Markov Model (HMM) profile (PF00891) was downloaded from the Pfam database to search the *C. sinensis* genome with an E-value < 1.0 × 10^−10^. To confirm that the results were error-free when screening out the proteins that did not belong to the target gene family, the protein sequences of these genes were also uploaded to PfamScan (https://www.ebi.ac.uk/Tools/pfa/pfamscan/ (accessed on 15 January 2022)) and SMART (http://smart.embl-heidelberg.de/ (accessed on 16 January 2022)), and an E-value < 1.0 × 10^−10^ was used for searching to ensure that any putative genes belonging to the *ASMT* gene family were not excluded from the analysis. Based on previous studies [25,27], the tea plant ASMT homologs with an amino acid homology higher than 31% were considered gene family members. In total, we obtained 20 candidate genes, and their chromosomal location information was used for numbering. The molecular weights and physicochemical properties of the tea plant ASMT proteins were analyzed using the online portal ExPASy (https://www.expasy.org/ (accessed on 19 January 2022)).

### 2.3. Phylogenetic and Structural Analyses of C. sinensis ASMT Genes

To determine the phylogenetic relationships of the tea plant *ASMT* genes, a phylogenetic tree was constructed using the ASMT protein sequences, implementing the neighbor-joining method in MEGA X software, along with bootstrap testing (n = 1000). Evolutionary trees were constructed using the Evolview online tool (https://evolgenius.info//evolview-v2 (accessed on 1 December 2022)).

### 2.4. Analysis of C. sinensis ASMT Motifs and Promoter Cis-Acting Elements

The genome information and gene structure annotation files of the tea plant were analyzed using TBtools 1.09876 [28], and the information files of the 20 candidate genes of the *ASMT* gene family were obtained and used to visualize the gene structure and to identify the number and arrangement of introns and exons. A motif analysis of the *ASMT* genes was performed using the MEME Suite online tool (http://meme-suite.org/index.html (accessed on 1 February 2022)) with the following parameters: the number of motifs = 10, occurrence per motif = 0 or 1, optimal motif width ranging from 6 to 50 amino acid residues, and maximum mismatch = 10. The 2.0 kb promoter sequence upstream of the ASMT proteins was extracted using TBtools and submitted to PlantCARE (http://bioinformatics.psb.ugent.be/webtools/plantcare/html/ (accessed on 1 February 2022)) in order to identify the active elements [29]. The results were mapped using TBtools software.

### 2.5. Chromosome Localization and Collinearity Analysis

The tea plant genome and chromosome annotation information were downloaded from the *C. sinensis* genome database (http://tpia.teaplant.org/index.html (accessed on 2 January 2022)), and the chromosomes were mapped using the GFF file configuration information. A collinearity analysis of the tea plant *ASMT* genes was performed using TBtools, while the Ka, Ks, and Ka/Ks ratios of the duplication events [30] were calculated using DNASP 6.0 software to further analyze the duration of duplication occurrences [31].

### 2.6. Expression Profiling of ASMT Family Genes in C. sinensis

The RNA-seq data were obtained from the Anhui Agricultural University *C. sinensis* genome database (http://tpia.teaplant.org/index.html (accessed on 3 February 2022)), and the gene expression levels were reported in fragment per kilobase of transcript per million mapped reads (FPKM). Expression profile mapping was performed using TBtools.

### 2.7. Quantitative Real-Time Fluorescence Analysis

A qRT-PCR analysis was performed on the cDNA of samples from different nodes of the experimental treatment using the Bio-Rad CFX96 (Bio-Rad, Hercules, CA, USA) fluorescence quantification system. Gene primers were first designed using Primer 5.0 software, with primer lengths of 150–250 bp (Appendix A), using the 1 μL cDNA template, 10 μL SYBR Premix ExTaq (Takara, Kyoto, Japan), 2 μL specific primers, and 7 μL ddH_2_O. The PCR thermal cycling parameters were as follows: 95 °C for 5 min, 45 cycles at 95 °C for 20 s, 60 °C for 20 s, and 72 °C for 10 s, 60 °C for 20 s, and 72 °C for 10 s for 45 cycles. CsPTB-RT is a specific primer for tea tree actin, and it was used as an internal control to normalize gene expressions. The relative expressions of genes were determined using the 2^−ΔΔCT^ method for analyses, and three biological replicates were performed for each sample.

## 3. Results

### 3.1. Genome-Wide Identification of ASMT Genes in C. sinensis

A total of 20 *ASMT* gene family members were identified in the tea plant genome via the bioinformatics analysis; they were sequentially named CsASMT01–CsASMT20 according to their position on the chromosomes. The gene name, gene ID, chromosomal position, open-reading frame (ORF), amino acid (AA), molecular weight (MW), and isoelectric point (pI) corresponding to each gene are listed in Table 1. The sequence analysis revealed that the *ASMT* gene family members had different corresponding ORFs, AAs, MWs, and pIs. The length of the ORFs ranged from 888 bp (CsASMT09) to 1182 bp (CsASMT18), and the length of the ASMT proteins ranged from 295 AAs (CsASMT11) to 393 AAs (CsASMT19). The MW of the proteins ranged from 32.28 (CsASMT11) to 43.80 kDa (CsASMT19), with a theoretical isoelectric point (pI) varying from 4.94 (CsASMT05) to 6.06 (CsASMT19). The isoelectric point analysis revealed that all the ASMT proteins were acidic.

### 3.2. Systematic Analysis and Conserved Motifs of C. sinensis ASMT Genes

For an in-depth understanding of the evolutionary relationships among the *ASMT* gene family members, a phylogenetic tree was constructed based on the 20 amino acid sequences of the tea plant (Figure 1). The results show that these ASMT proteins could be divided into three subfamilies, which were classified as subfamilies I, II, and III. The largest subfamily, subfamily III, contained nine ASMT members, followed by subfamilies I and II, which contained seven and four ASMT members, respectively. Members of the same subfamily showed a high level of homology.

To better characterize the diversity of the CsASMT proteins, the AA sequences of the 20 genes were analyzed using the MEME online tool, revealing a total of 10 different motifs, namely, motifs 1–10 (Figure 2), whose corresponding protein sequences ranged from 15 to 50 AAs in length. The distribution of most of the gene motifs was relatively concentrated. Motifs 1, 2, and 4 were present in all genes (Figure 2), indicating that these proteins were highly conserved during the evolution of the tea plant. However, CsASMT11 was unique because it did not contain motifs 5, 7, or 9, which were present in most genes. Similarly, individual genes showed minor differences in motif distribution when comparing subclade members or members of the same subclade branch, which indicates that the *ASMT* genes in tea plants underwent different selection pressures and evolutionary processes. Our gene structure analysis showed that all CsASMT genes contained introns; 20% of them contained three introns, while the rest contained one or two introns. The comparison also revealed that the genes with a higher number of introns had a relatively lower motif number, which confirmed the previous conclusion that the number and length of introns may be related to the loss of conserved motifs as a result of evolutionary processes.

### 3.3. Chromosome Distribution and Collinearity Analysis

The chromosomal localization analysis of the ASMT gene family members revealed that the 20 genes were located on seven chromosomes (Figure 3). Five *CsASMT* genes were distributed on chromosomes 5 and 7, whereas only one was detected on chromosomes 1 and 2. The collinearity analysis showed that only two pairs of the 20 *ASMT* genes, namely, *CsASMT03/CsASMT13* and *CsASMT10/CsASMT17*, had a linear relationship (Figure 4). Thus, we concluded that the ASMT gene family expansion occurred via fragment replication. In the analysis of the replication event time, the Ka/Ks ratios of the two gene pairs were lower than 1, and these genes suffered from an evolutionary negative purifying selection over 19.78 to 34.66 million years (Table 2).

### 3.4. Analysis of Cis-Acting Elements of the C. sinensis ASMT Promoter

An analysis of cis-acting elements is important for predicting the regulation of gene expressions and for understanding the possible involvement of genes in the related regulation pathways. In the present study, by investigating the 2000 bp promoter region upstream of the ASMT gene family members, three types of cis-acting elements were found: those associated with stress responses, those associated with hormone responses, and those associated with plant development. Based on the predicted results, the promoters were classified into 21 main types of promoter cis-acting elements, including those associated with cell cycle regulation, light responses, the MYBHv1 binding site, low-temperature responses, gibberellin responses, abscisic acid responses, and drought induction (Figure 5). Moreover, based on a functional classification, antioxidant cis-acting elements (ARE) were found in most of the promoter regions of the *CsASMT* genes. In addition, stress-response elements, including low-temperature-response elements, defense-response elements (TC-rich repeats), and drought-response elements (MBS), were distributed in the gene promoters. Most genes contained abscisic-acid-responsive elements (ABRE) and MeJA-responsive elements (TGACG-motif and CGTCA-motif), which are hormone-responsive elements, and these cis-acting elements were widely distributed and numerous. Among the cis-elements associated with plant development, most genes contained cis-acting elements related to light responses and soluble protein metabolism, such as Box 4, O2-site, and G-Box (Figure 6). Therefore, *ASMT* genes may play important roles in the response of tea plants to environmental stresses.

### 3.5. Expression Profiles of CsASMT Genes under Different Types of Stress

The FPKM values obtained from the RNA-seq data were used to construct two heat maps of the 20 *CsASMT* genes under cold and drought stresses in order to explore the expression patterns of the ASMT gene family in tea plants under different stress conditions. The results show that eight genes were not expressed before or after treatment under cold stress conditions. Moreover, according to the RNA-seq analysis results, most of the expressed *CsASMT* genes showed differential expression levels after 6 h of treatment, and their expression levels were not high (Figure 7). We identified seven differentially expressed genes, namely, *CsASMT01*, *CsASMT08*, *CsASMT09*, *CsASMT10*, *CsASMT18*, *CsASMT19*, and *CsASMT20*, by combining the expression results after 7 d of treatment. The recovery culture showed high expression levels of these genes and relatively stable replicate results. Among them, the expression levels of *CsASMT01* and *CsASMT20* were consistently maintained, and these genes were differentially expressed before and after stress treatment. The expressions of *CsASMT01* and *CsASMT18* showed significant increases under cold stress, whereas the expression of *CsASMT20* decreased under cold stress; however, it returned to a normal level after the recovery culture. The expressions of all other genes decreased after treatment, and the recovery culture expression did not reach a normal level. These results suggest that the seven *CsASMT* genes may mediate cold damage in tea plants. Similarly, in the construction of the heat map of the 20 *CsASMT* genes under drought stress, we found that the expressions of *CsASMT08*, *CsASMT10*, *CsASMT18*, *CsASMT19*, and *CsASMT20* decreased significantly after drought treatment and during each of its three stages (Figure 8); *CsASMT01* and *CsASMT20* were highly expressed before and after treatment, and the expression of *CsASMT01* increased after drought treatment.

To verify the RNA-seq results and to further analyze which of these genes had a mediating effect under drought treatment versus low-temperature treatment, we further validated the seven selected differentially expressed genes using a qRT-PCR analysis (Appendix A). Under drought treatment, *CsASMT08*, *CsASMT09*, and *CsASMT10* were negatively regulated at different treatment time points, and *CsASMT18* and *CsASMT20* were upregulated at 0.5 d and 1 d of treatment, respectively; the other genes were not significantly different before and after treatment. Under low-temperature treatment, *CsASMT08* and *CsASMT10* were significantly upregulated at different time points of treatment; *CsASMT20* was also significantly upregulated at 1 d and 4 d of treatment (Figure 9); *CsASMT01*, *CsASM09,* and *CsASMT19* showed negative regulation at different treatment time points; and *CsASMT18* showed no significant difference before and after treatment. These results also indicate that *CsASMT08*, *CsASMT09*, *CsASMT10*, and *CsASMT20* may play vital roles in responses to environmental stresses in the tea plant.

## 4. Discussion

Melatonin is widely known as a multifunctional molecule, and it plays an important role in oxidative stress tolerance in plants [32]. Therefore, melatonin-related studies are a crucial part of plant adversity research, and studying the function of ASMT is important, as it is one of the key enzymes participating in melatonin synthesis in plants [23]. Previous studies have shown that exogenous melatonin applications can improve the ability of plants to cope with abiotic stresses, and the functional validation of the related genes has been demonstrated in rice and *A. thaliana* [24,25]. In the present study, 30 candidate *ASMT* genes were initially identified through an analysis of the tea plant genome. Considering that multiple putative ASMT gene family members have been identified in rice, 20 *ASMT* genes were identified with a rice ASMT homology greater than 31% with structural proteins, while 10 gene family members with a low level of homology (less than 31%) were excluded. The ASMT gene family in tea plant is larger than that in tomato (14) and pepper (16); however, it contains fewer *ASMT* genes than the gene families of mulberry (20) and apple (37), indicating that it evolved differently in different species.

To explain the evolutionary relationships between the *ASMT* family genes in the tea plant, we constructed a phylogenetic tree using aligned protein sequences. Our cluster analysis showed that the 20 tea plant *ASMT* genes could be classified into three subfamilies with six homologous gene pairs (*CsASMT16*/*CsASMT19*, *CsASMT15*/*CsASMT18*, *CsASMT07*/*CsASMT09*, *CsASMT10*/*CsASMT11*, *CsASMT12*/*CsASMT14*, and *CsASMT04*/*06*), which were relatively conserved and perhaps functionally similar. To further investigate the structural functions of these *ASMT* genes, we performed a motif analysis and found that the ASMT members in the same subfamily or cluster had similar motif compositions, further supporting the subclade classification identified in the phylogenetic analysis. The tea plant ASMT gene family structure was also found to be highly conserved according to the motif distribution. An exon–intron organization analysis showed that all the genes contained introns, which were relatively long and interrupted the gene coding sequence. If a gene has fewer introns, it may respond to external stress regulation more rapidly; however, highly expressed plant genes generally contain more introns than lowly expressed genes [33], which indicates that *ASMT* genes may play roles in abiotic stress responses.

Gene duplication is the basis of gene diversity, leading to genetic novelty; therefore, it is the main source of species evolution and adaptation [34,35]. It is divided into tandem and fragment duplications; it is also one of the main factors in the evolution of gene families, and it contributes to differences in gene size and distribution [36,37]. Our analysis revealed that only two pairs of tea plant *ASMT* genes underwent fragment duplication and that none underwent tandem duplication; this low frequency of duplication may be the underlying reason for the similar lengths of the *ASMT* genes. Furthermore, the occurrence of duplication events indirectly indicated that genetic information was being transmitted, which also suggests that fragment duplication may have played an important role in the evolution of the tea plant *ASMT* genes. The 20 tea plant *ASMT* genes were unevenly distributed on seven chromosomes and were mostly located in the middle of the chromosome; this may explain why most of them were not involved in the duplication events. Similarly, tandem repeats were found in the *ASMT* genes of the pepper plant.

The gene expression patterns of plants under various stresses can provide important clues for gene function analyses, and cis-acting element analyses can help predict the network of regulatory mechanisms in which genes may be involved. In the present study, we analyzed the expression profiles of the *ASMT* genes in the tea plants under drought and low-temperature stress conditions using RNA-seq and qRT-PCR. The results of the RNA-seq analysis showed that eight genes were not expressed before or after treatment under cold stress conditions, and the expression analysis comparing 6 h of treatment, 7 d of treatment, and after the recovery culture revealed that *CsASMT01 CsASMT08*, *CsASMT09*, *CsASMT10*, *CsASMT18*, *CsASMT19,* and *CsASMT20* responded to cold stress and drought stress. To further identify which of these *CsASMT* genes are most important in mediating cold and drought stresses, these seven differentially expressed *CsASMT* genes were further validated using qRT-PCR. The results show that *CsASMT08*, *CsASMT09*, and *CsASMT10* were negatively regulated to drought and that *CsASMT01* and *CsASMT20* were positively regulated to drought; *CsASMT08*, *CsASMT10*, and *CsASMT20* positively responded to low-temperature stress, and *CsASM09* negatively fed back to low-temperature stress, which indicates that these genes may have different response mechanisms to drought and low temperatures and that they may play central roles in abiotic stress processes. Similarly, in grapes, it has been found that the overexpression of *VvASMT1* enhances tolerance to salt and osmotic stresses in *Nicotiana benthamiana* [38]. These results suggest that *ASMT* genes play important roles in abiotic stress processes in plants. It is well-known that cis-acting elements, ABREs, MBSs, G-boxes, AREs, and TC-rich repeats play important roles in drought stress responses and the regulation of downstream gene expressions [39,40,41]. The cis-acting element analysis revealed that *CsASMT08*, *CsASMT09*, *CsASMT10,* and *CsASMT20* contained one to five stress-like cis-acting elements; for example, *CsASMT08* contained one ARE, and *CsASMT10* contained one ARE and four ABREs, indicating that all these genes may mediate the stress response pathways.

## 5. Conclusions

In our study, we identified 20 *ASMT* genes by bioinformatics methods and analyzed them in terms of phylogeny, gene structure, and cis-acting elements to provide a preliminary resolution and better understanding of the ASMT gene family in tea plant. We showed that the expressions of *CsASMT08, CsASMT09, CsASMT10,* and *CsASMT20* were differentially induced under drought and low-temperature treatments, indicating that they may regulate abiotic stress resistance in the tea plant. These results lay the foundation for further studies on the functions of *ASMT* genes, as well as for identifying candidate genes for abiotic stress resistance regulation in tea plants.

## Figures and Tables

**Figure 1 genes-14-00409-f001:**
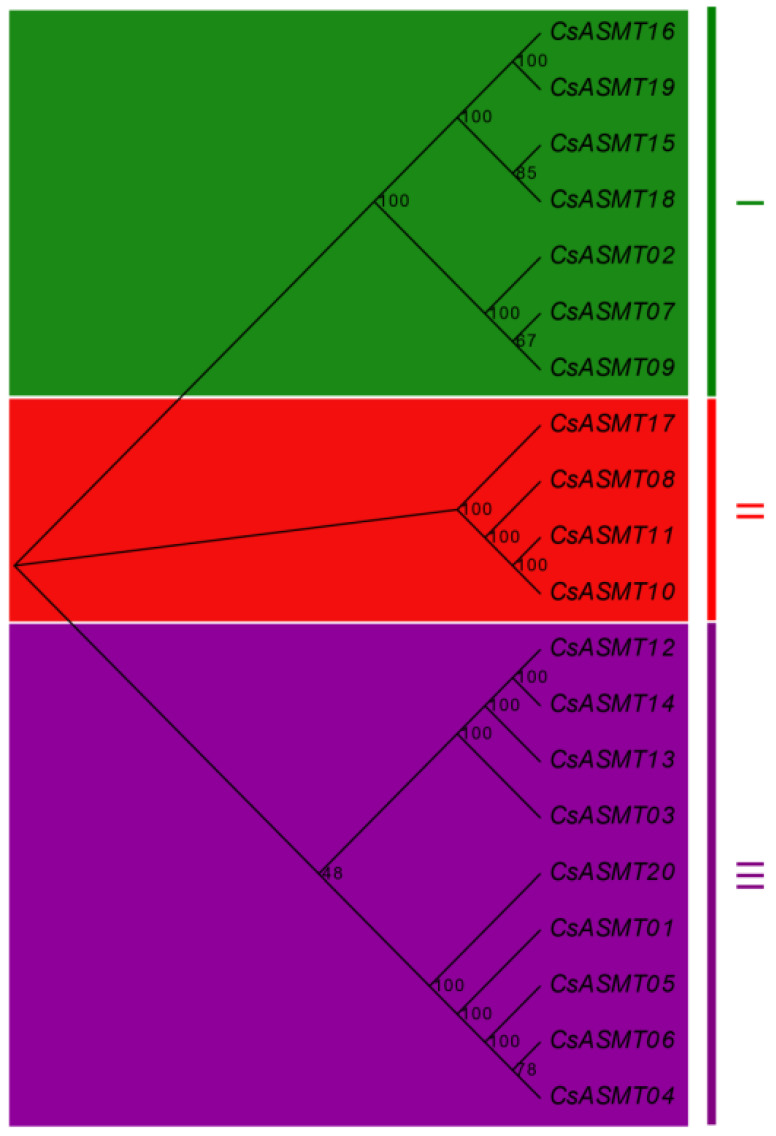
Phylogenetic analysis of *ASMT* genes in *C. sinensis.* The phylogenetic tree was constructed based on the sequences of CsASMT proteins using the neighbor-joining method with bootstrap analysis (1000 replicates) in MEGA X.

**Figure 2 genes-14-00409-f002:**
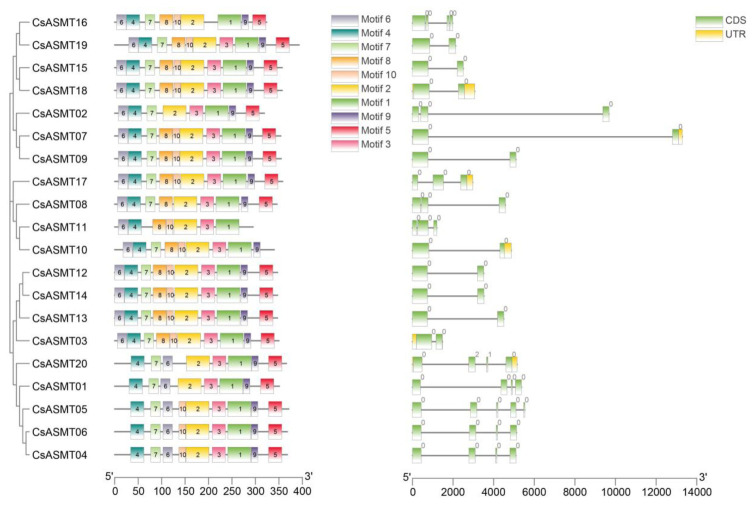
Conserved motifs and gene structure analysis of *C. sinensis* ASMT gene family. Different colored boxes with numbers on the left side represent different types of motifs. UTRs are represented by green boxes on the right side of the figure, CDS sequences are represented by yellow rectangles and introns are represented by grey lines.

**Figure 3 genes-14-00409-f003:**
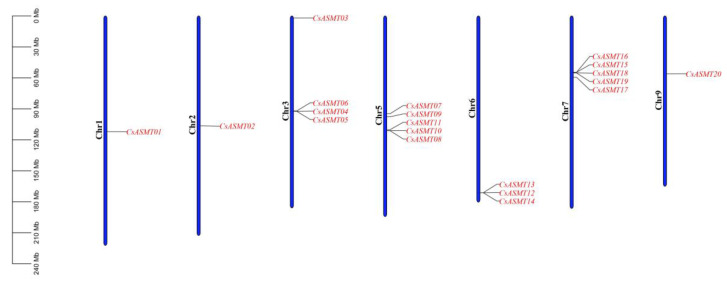
Chromosome localization analysis of *C. sinensis* ASMT gene family.

**Figure 4 genes-14-00409-f004:**
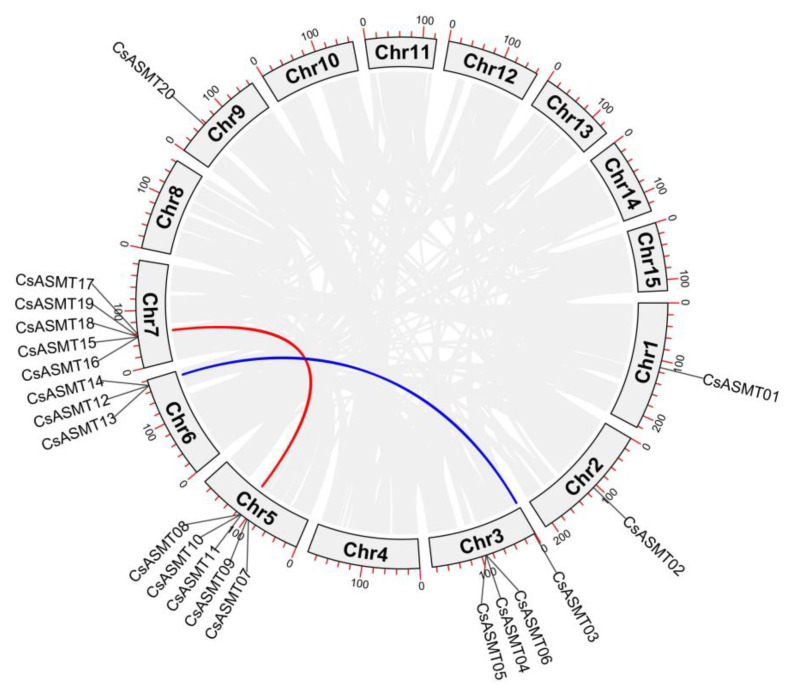
Collinearity analysis of *C. sinensis* ASMT gene family. The lines between chromosomes indicate a fragmentary replication relationship between *ASMT* genes in tea plant.

**Figure 5 genes-14-00409-f005:**
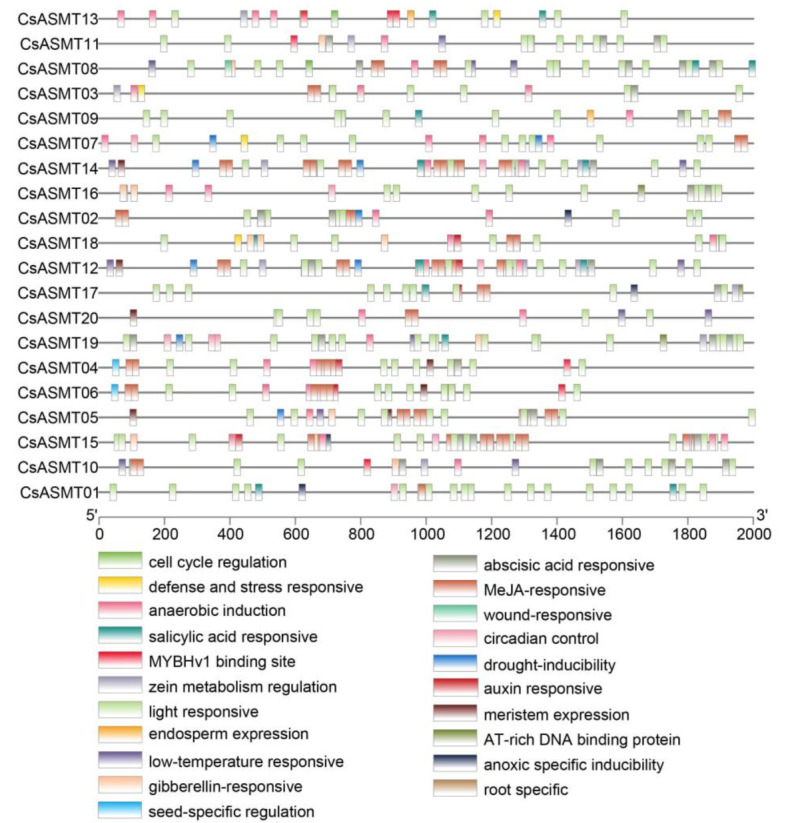
Cis-element analysis of *C. sinensis ASMT* gene promoters. The binding sites of promoter regions are indicated by the different colored boxes.

**Figure 6 genes-14-00409-f006:**
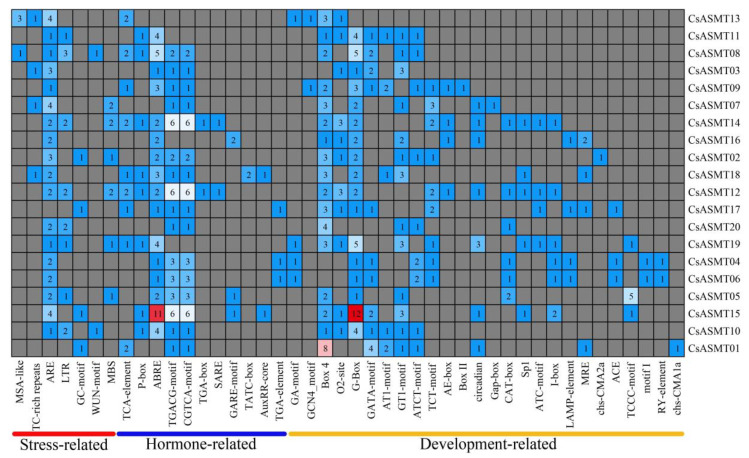
Distribution of the number of cis-acting elements in the *C. sinensis ASMT* gene promoter.

**Figure 7 genes-14-00409-f007:**
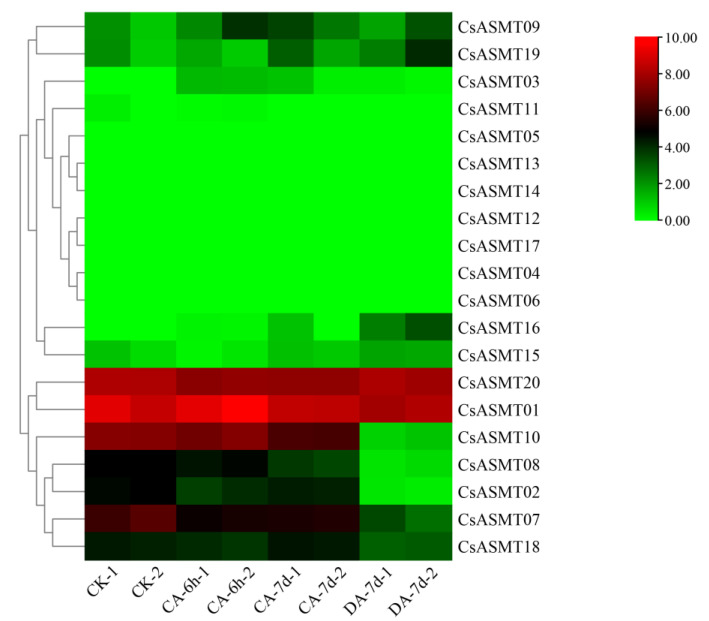
Heat map of FPKM expressions of *C. sinensis ASMT* genes under cold stress based on RNA-Seq data. FPKM values were all log2-processed, and heat map clusters were constructed. The color scale representing relative expression values is shown on the left, CA: 10 °C treatment, DA: room-temperature recovery.

**Figure 8 genes-14-00409-f008:**
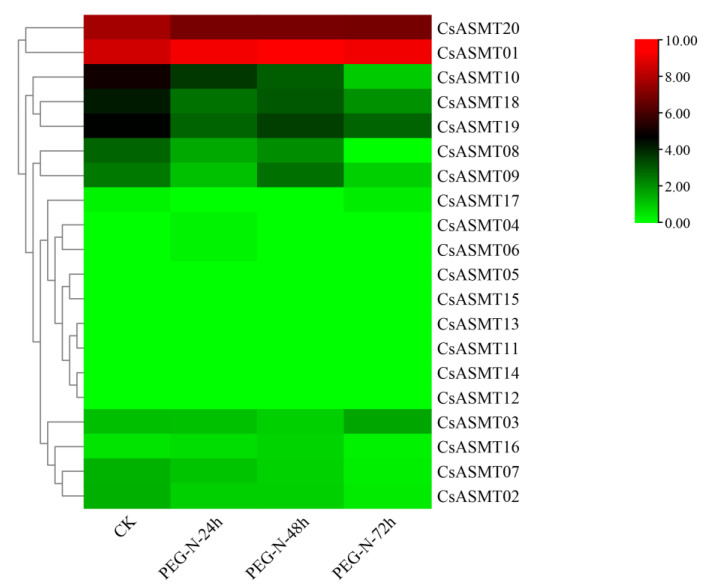
Heat map of FPKM expressions of *C. sinensis ASMT* genes under drought stress based on RNA-Seq data. FPKM values were all log2-processed, and heat map clusters were constructed. The color scale representing relative expression values is shown on the left.

**Figure 9 genes-14-00409-f009:**
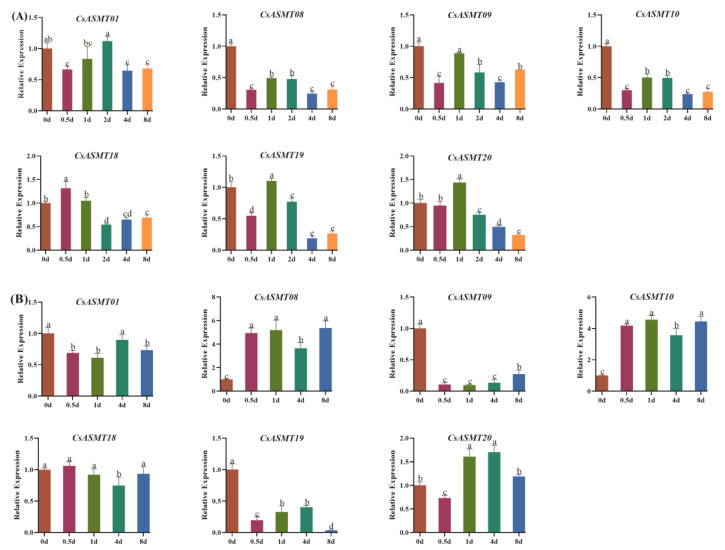
(**A**) qRT-PCR of 7 selected *C. sinensis ASMT* genes in response to drought stress; (**B**) qRT-PCR of 7 selected *C. sinensis ASMT* genes in response to cold stress. Bars represent the mean ± standard deviation of three replicates, different letters indicate significant differences at different time points, and all data were subjected to one-way ANOVA and LSD test (*p <* 0.05).

**Table 1 genes-14-00409-t001:** Physical and chemical properties of ASMT proteins in *C. sinensis*.

Gene Name	Gene ID	Group	Chr	Location	AA	MW	pI	ORF
*CsASMT01*	CSS0048493	III	Chr1	112085937-112091308	351	38.32	5.94	1056
*CsASMT02*	CSS0025746	I	Chr2	106494711-106504385	319	34.95	5.64	960
*CsASMT03*	CSS0006416	III	Chr3	1899577-1901052	350	38.66	5.49	1107
*CsASMT04*	CSS0037320	III	Chr3	92351303-92356400	368	39.85	5.14	1107
*CsASMT05*	CSS0039718	III	Chr3	92432006-92437542	371	40.17	4.94	1116
*CsASMT06*	CSS0037946	III	Chr3	91956040-91961165	368	39.87	5.14	1053
*CsASMT07*	CSS0007434	I	Chr5	94698511-94711799	354	39.10	5.45	1065
*CsASMT08*	CSS0003573	II	Chr5	111181605-111186182	346	38.95	5.87	1068
*CsASMT09*	CSS0006936	I	Chr5	97659841-97664938	355	39.16	5.6	888
*CsASMT10*	CSS0044619	II	Chr5	111082799-111087666	340	38.50	5.88	1023
*CsASMT11*	CSS0003281	II	Chr5	110166060-110167265	295	32.28	5.24	1041
*CsASMT12*	CSS0033599	III	Chr6	171235170-171238679	347	38.85	5.47	1044
*CsASMT13*	CSS0000080	III	Chr6	171195877-171200372	347	38.88	5.47	1044
*CsASMT14*	CSS0015347	III	Chr6	171279383-171282903	347	38.85	5.47	1044
*CsASMT15*	CSS0042551	I	Chr7	54550266-54552772	357	39.62	5.57	975
*CsASMT16*	CSS0023656	I	Chr7	54505207-54507188	324	36.28	6	1074
*CsASMT17*	CSS0035451	II	Chr7	59409307-59412279	358	40.53	5.55	1074
*CsASMT18*	CSS0029228	I	Chr7	55144050-55147115	357	39.59	5.57	1182
*CsASMT19*	CSS0036567	I	Chr7	55209520-55211648	393	43.80	6.06	1077
*CsASMT20*	CSS0036494	III	Chr9	56007566-56012718	366	40.16	5.95	1101

**Table 2 genes-14-00409-t002:** The parameters and data of the duplication evens in the CsASMTs.

Gene Name	Gene Name	Ka	Ks	Ka/Ks	Date (Million Years)
*CsASMT03*	*CsASMT13*	0.13	0.59	0.23	17.89
*CsASMT10*	*CsASMT17*	0.22	1.04	0.21	34.66

Note: the synonymous substitution rate (Ks) and the non-synonymous substitution rate (Ka) were calculated using the program DNASP 6.0, T = Ks/2λ is the formula used to calculate the date of the repeated event, λ is the estimated clock sample rate of synonymous substitution in dicotyledons, and λ = 1.5 × 10^−8^ substitutions/synonymous sites/year.

## Data Availability

Data available within the article or its Appendix A. The authors confirm that the data supporting the findings of this study are available within the Appendix A.

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
