# Peer review of "Genome Identification of the Tea Plant (Camellia sinensis) ASMT Gene Family and Its Expression Analysis under Abiotic Stress"

_genes, 2023, doi:10.3390/genes14020409_

Round 1

Reviewer 1 Report

Figure 9 should be improved. It is not visible.

Reviewer 2 Report

The topic of the research is interesting and attractive since climate change impose severe threats on agriculture. in this context it is important to improve the knowledge upon the underlying mechanisms of stress tolerance in plants especially in agricultural crops. The research is well designed however some important details are missing (see precise observations in the pdf attached). The genetic part is very complex, complete and clearly presented while the presentation of the biological material, treatments need improvement so the Conclusions.
